# Should the Definition of Low Birth Weight Be Same in Every Ethnicity Considering the DOHaD Concept?

**DOI:** 10.3390/pediatric17010008

**Published:** 2025-01-16

**Authors:** Yoshifumi Kasuga, Mamoru Tanaka

**Affiliations:** Department of Obstetrics and Gynecology, Keio University School of Medicine, 5 Shinanomachi, Shinjuku-ku, Tokyo 160-8582, Japan; mtanaka@keio.jp

**Keywords:** low birth weight, developmental origins of health and disease, pregnancy, ethnicity

## Abstract

Low birth weight (LBW) is a significant concern not only because of its association with perinatal outcomes, but also because of its long-term impact on future health. Despite the physical differences among individuals of different ethnicities, the definition of LBW remains the same for all ethnicities. This study aimed to explore and discuss this issue. We compiled national data from several countries and found that maternal height was negatively correlated with LBW incidence. We discovered the INTERGROWTH-21st chart may not be suitable for the Japanese population, as the Japanese birth weight chart differs from the INTERGROWTH-21st chart. Researchers have reported different LBW cutoff values used to assess adverse perinatal outcomes for different countries. However, there is currently no definition of LBW independent of the mother’s country of origin that can be used for predicting the risk of adverse health outcomes. Therefore, the current era of personalized healthcare may be the perfect time to establish a standard definition of LBW which is independent of the mother’s country of origin. Considering the future of healthcare, it seems an apt time to discuss the development of a more meaningful definition of LBW that can be applied across ethnicities. Further research is needed to investigate the cutoff values of LBW in every ethnicity.

## 1. Introduction

In 2004, the World Health Organization (WHO) defined low birth weight (LBW) as weighing < 2500 g at birth. Neonates with LBW are at a high risk of perinatal complications and poor health. The developmental origins of health and disease hypothesis considers LBW to be one of the most important problems related to future healthcare. Clinicians should strive to reduce LBW births through preconception and perinatal care. However, although each ethnic group is associated with a different physique, the definition of LBW is the same for every ethnicity. Therefore, two questions arise from assessing the adverse health risks associated with LBW: 1. Are there differences in the health risks of LBW infants among mothers from different countries? Does the birth weight cutoff value based on the mother’s country of origin increase the risk of adverse health effects? 2. Should we consider the causes of LBW when evaluating its risk? When we successfully predicted future adverse outcomes (e.g., diabetes, hypertension) for the next generation, we considered that the definition of LBW might need to be reconsidered.

With this in mind, we discuss the definition of LBW and its implications for future health in Japan.

## 2. Causes of LBW for Countries and Environments

According to several previous reports, the causes of LBW vary across countries and environments, including meta-analyses. Maternal infections may be associated with LBW. For example, maternal malaria and dengue virus infections increase the risk of LBW [1,2]. Maternal HIV infection was associated with a higher risk of LBW in a meta-analysis of eight countries [3]. Exposure to particulate matter with an aerodynamic diameter of ≤2.5 μm (PM_2.5_) has been linked to LBW [4]. In Brazil, not only PM_2.5_ but also nitrogen dioxide (NO_2_) and ozone (O_3_) were found to have robust associations with LBW, depending on the area [5]. According to Australian data, flood exposure in the 13–24 weeks before the last menstrual period significantly increases the risk of LBW in full-term births [6]. However, earthquakes were not found to be associated with LBW in a meta-analysis of 13 studies [7]. In Nepal, drought during the first trimester and excess rainfall during the third trimester are associated with LBW [8]. However, there are gaps in the literature regarding the association between climate change, extreme weather, and adverse perinatal outcomes. Even if mothers are affected by the same exposure, maternal mental health issues may not manifest in the same manner [9]. Maternal exposure and risk factors for LBW differ across countries, meaning that the causes of LBW are not uniform. Further research is needed to investigate the long-term adverse outcomes associated with LBW.

## 3. Is LBW a Risk Factor for Future Health in Every Ethnicity, Considering the Developmental Origins of Health and Disease (DOHaD) Concept?

Ethnicity may be the most important factor for fetal growth. However, although maternal height was correlated with LBW incidence in 19-year-olds categorized by country (Figure 1) [10], LBW is defined by birth weight and not maternal diversity. Voigt et al. reported that maternal height and weight were associated with offspring birth weight, and big mothers had lower rates of children who were small for gestational age (SGA: birth weight < 10%tile) and higher rates of those who were large for gestational age (LGA: birth weight ≥ 90%tile) [11]. As maternal height is a good predictor of birth weight and the average maternal height is not the same in every ethnicity, we believe that the definition of LBW should differ and be based on maternal physique and ethnicity. However, Rochow et al. examined whether maternal physique and ethnicity were more predictive of LBW and reported that maternal height was a stronger predictor of birth weight than ethnicity [12]. Notably, because Rochow et al. included women of various ethnicities living in Germany, their nutritional statuses may have been the same. Moreover, when considering the association between ethnicity and LBW, whether the mothers are living in their original country (e.g., Japanese in Japan, German in Germany, or Chinese in China) should also be evaluated. This is because, compared with Caucasians and Hispanics, Asians are more likely to have beta cell dysfunction and impaired insulin secretion [13,14]. Additionally, a diet suitable for Caucasians and Hispanics may provide excess energy to Asians. Therefore, being Asian in Europe and the US has been reported as a risk factor for gestational diabetes (GDM) [15,16]. However, the incidence of GDM of approximately 10% among Japanese people living in Japan is not much higher than that among Caucasians [17]. This might mean that Japanese food and the Japanese lifestyle may control maternal healthcare. Moreover, human leukocyte antigen (HLA) type might affect fetal growth. Emmery et al. reported that HLA-G variations were associated with fetal weight and placental weight [18]. Furthermore, the INTERGROWTH-21st project revealed that race and ethnicity did not influence fetal growth. However, the sample population included people from only eight countries (Brazil, China, India, Italy, Kenya, Oman, the United Kingdom, and the United States), and the sample size was small [19], excluding countries with a high LBW incidence. If this study were performed in other countries, this result might have changed. In fact, for Chinese non-high-risk babies, the SGA and LGA rates calculated using the Chinese chart differed from those calculated using the INTERGROWTH-21st chart. The rate of SGA using the Chinese chart was 10.1%, whereas that using the INTERGROWTH-21st chart was 6.5%. Similarly, the rate of LGA using the Chinese chart was 9.9%, compared to 8.2% when using the INTERGROWTH-21st chart [20]. We compared the centile curves for birth weight between Japanese newborns and those from the INTERGROWTH-21st study (Figure 2). Additionally, the INTERGROWTH-21st chart may not be suitable for the Japanese population. This is because the Japanese birth weight chart differs from the INTERGROWTH-21st chart [21,22]. Furthermore, Dola et al. used the 2022 Centers for Disease Control and Prevention’s National Natality Dataset to predict LBW based on several factors, including anthropometric, socioeconomic, and demographic data from parents. They concluded that the threshold for LBW in Asians and Hispanics may be lower than the WHO definition (<2500 g) [23].

Researchers have reported different LBW cutoff values for different countries that are used to assess adverse perinatal outcomes. For example, 2750 g was used in the US in 1922 and 3000 g in Denmark in 2007. Weighing under 2200 g could serve as an appropriate definition for LBW, considering the risks of early neonatal mortality among African, Latin American, and Asian populations [24]. However, there is currently no LBW definition independent of the mother’s country of origin that is useful for predicting the risk of adverse health outcomes. Given the higher incidence of LBW, many reports on the future adverse health risks associated with LBW should be published in low- or middle-income countries. On the other hand, there is a gap between perceived risk and sense of urgency, as highlighted by the number of reports in a previous review [25]. Therefore, we believe that data from both high- and low-income or middle-income countries should be investigated and discussed.

## 4. DOHaD Related to LBW in Japan

In Japan, the incidence of LBW at term is higher than that in other developed countries, despite the lower rate of preterm birth [26]. The reasons for the higher incidence include younger age, nulliparity, pre-pregnancy underweight (BMI < 18.5 kg/m^2^), inadequate gestational weight gain (GWG), birth at 37 gestational weeks, delivery via cesarean section, hypertensive disorder of pregnancy (HDP), smoking, female neonate, and neonatal congenital anomaly [26]. Furthermore, maternal heated tobacco use is associated with LBW and HDP [27]. A meta-analysis of five ongoing prospective birth cohort studies reported an increased risk of SGA with pre-pregnancy underweight, inadequate GWG, and smoking [28]. Maternal underweight is an important issue in Japan [29]. Therefore, preconception care regarding nutritional habits is important to improve the future health care of offspring. Regarding perinatal outcomes, mothers who were born as LBW infants are at a higher risk of delivering LBW or SGA infants [30]. Furthermore, both maternal and paternal LBW are associated with LBW delivery [31]. Therefore, impaired parental fetal growth may influence fetal growth in the offspring. Lower maternal birth weight has an increased risk of perinatal complications (e.g., HDP, GDM, and preterm delivery) [32,33]. Among 3107 participants (2303 men and 804 women), LBW was independently associated with adult hypertension in a Japanese workplace population [34]. In adolescence, the rate of stage 2 chronic kidney disease is significantly related to LBW [35]. Japanese high school girls born with LBW have significantly higher systolic and diastolic blood pressures, triglyceride levels, insulin levels, and insulin resistance than those whose birth weights are ≥3400 g [36]. LBW is associated with high low-density lipoprotein and total cholesterol levels in men and hypertension and diabetes mellitus in women aged 40–69 years [37]. Therefore, in Japanese individuals, birth weight < 2500 g may be a good predictor of adverse health, considering the DOHaD concept. However, LBW encompasses preterm, term, extreme LBW (<1000 g), very LBW (<1500 g), multifetal pregnancy, and infants with congenital anomalies. Furthermore, very low birth weight was associated with a higher prevalence of cardiovascular disease, hypertension, and diabetes when babies weighing 3000–3999 g were used as the reference, although babies weighing 1500–2499 g showed only a weak association [38]. The DOHaD holds that preconception, prenatal, and early postnatal environments affect health outcomes in childhood and adulthood. Furthermore, LBW incidence is higher in Japan than in other developed countries, despite the lower rate of preterm births. However, many previous studies did not consider the causes of LBW when assessing the associated health risks; some reports were retrospective in design, and some had small sample sizes. As no prospective analysis has been conducted to determine the Japanese cutoff value for LBW, considering the causes of LBW, further research is required to establish an appropriate cutoff birth weight.

## 5. Conclusions

Recently, personalized medicine has been applied to various diseases, with some malignancies selected for treatment based on patient genetic information. The incidence of preeclampsia in women and poor childhood growth in LBW infants differs according to ethnicity and race [39]. Therefore, ethnic and racial diversity should be considered when discussing the adverse health risks of LBW.

In conclusion, the current era of personalized healthcare may be the perfect time to establish a standard definition of LBW, independent of the mother’s country of origin. Considering future healthcare, it seems timely to discuss the development of a more meaningful definition of LBW that can be applied across ethnicities. Therefore, we conclude that further research is needed to investigate the cutoff values of LBW in every ethnicity.

## Figures and Tables

**Figure 1 pediatrrep-17-00008-f001:**
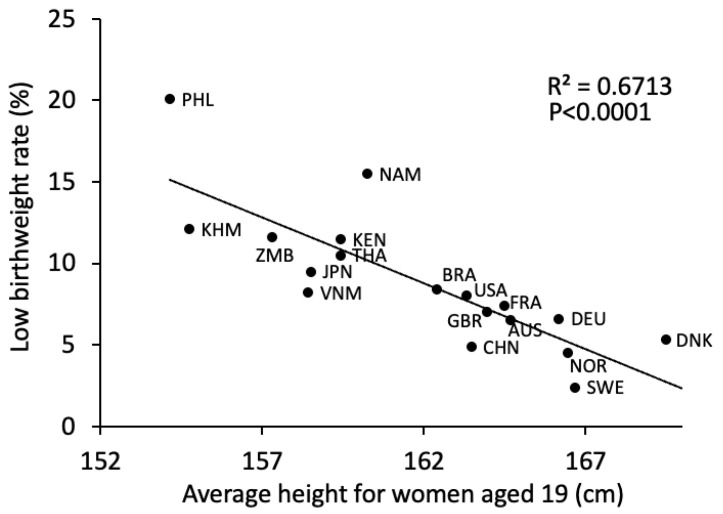
Correlation between low birth weight rate and average height in women aged 19 years in 18 countries. An inverse relationship was noted between average height in women aged 19 years and low birth weight rate (r^2^ = 0.6713, *p* < 0.0001).

**Figure 2 pediatrrep-17-00008-f002:**
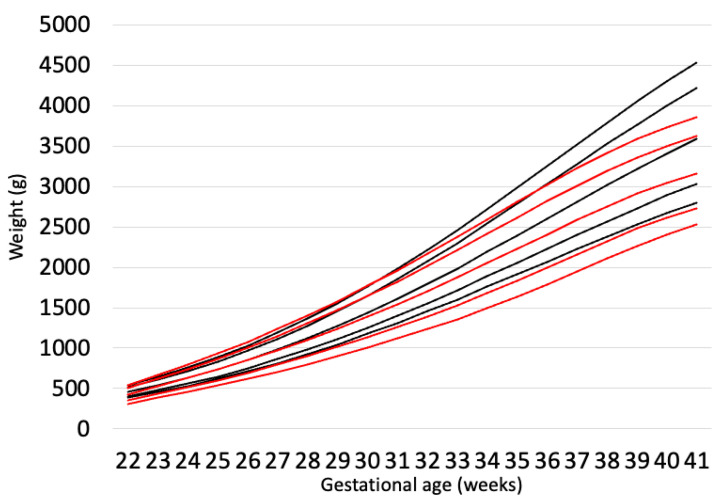
Comparison of centile curves for birth weight between Japanese newborns (red line) and INTERGROWTH-21st newborns (black line).

## Data Availability

No new data were created or analyzed in this study. Data sharing is not applicable to this article.

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
