# Peer review of "Should the Definition of Low Birth Weight Be Same in Every Ethnicity Considering the DOHaD Concept?"

_pediatrrep, 2025, doi:10.3390/pediatric17010008_

Round 1

Reviewer 1 Report

Comments and Suggestions for Authors

The authors pointed out an important but rarely discussed topic in LBW research. This opinion piece does point out the critical problems that need to be resolved. One key issue in studying birth weight is using the registered birth weight without input from maternal medical information such as blood pressure, blood sugar, BMI, smoking, drug use, infections, etc. Ideal population birth weight analysis should exclude data from babies born to mothers with significant medical conditions. Unfortunately, such a study is unavailable. I could not find Figure 1 in the main text, and I am interested in knowing where the figure is coming from. I could not find the figure from Rochow’s original article.

You raised two questions in the introduction, but there were three instead (lines 20-24). I think ethnicity (race) should be used instead of country to discuss the definition of LBW (lines 18-20). It is well-perceived that Black neonates have a higher percentage of LBW than other ethnic groups in the USA, and there is a wide variation in the percentages of LBW in different regions of China. The content in Section 2 needs rearrangement to avoid conflicting information. There are two section 3s but no section 4. The 1st sentence of the first section 3 (lines 49-50) is conceptually wrong as the rate of LBW is plotted against country instead of ethnicity. The higher percentage of glucose intolerance during pregnancy for Asian women seems to contradict the finding that LBW is more common in Asian ethnic groups. The information in this section is not well-organized, and positive and negative findings mingle. This might cover all aspects from previous studies but only confuses readers.  In the second section 3, line 117, it is described that “mothers who were born with LBW” should be “mothers who were born as LBW.” 

Comments on the Quality of English Language

If the manuscript were better edited, it would be easier for the readers to comprehend. There are numerous minor grammatical errors and inappropriate word use. 

Author Response

Reviewer 1

The authors pointed out an important but rarely discussed topic in LBW research. This opinion piece does point out the critical problems that need to be resolved. One key issue in studying birth weight is using the registered birth weight without input from maternal medical information such as blood pressure, blood sugar, BMI, smoking, drug use, infections, etc. Ideal population birth weight analysis should exclude data from babies born to mothers with significant medical conditions. Unfortunately, such a study is unavailable. I could not find Figure 1 in the main text, and I am interested in knowing where the figure is coming from. I could not find the figure from Rochow’s original article.

 [Response]

Thank you for your comment. I wish to share the importance of this issue with you. Figure 1 can be found at the top of page 3.

You raised two questions in the introduction, but there were three instead (lines 20-24). I think ethnicity (race) should be used instead of country to discuss the definition of LBW (lines 18-20). It is well-perceived that Black neonates have a higher percentage of LBW than other ethnic groups in the USA, and there is a wide variation in the percentages of LBW in different regions of China. The content in Section 2 needs rearrangement to avoid conflicting information. There are two section 3s but no section 4. The 1st sentence of the first section 3 (lines 49-50) is conceptually wrong as the rate of LBW is plotted against country instead of ethnicity. The higher percentage of glucose intolerance during pregnancy for Asian women seems to contradict the finding that LBW is more common in Asian ethnic groups. The information in this section is not well-organized, and positive and negative findings mingle. This might cover all aspects from previous studies but only confuses readers.  

[Response]

Thank you for your editorial advice and comment. I agree that ethnicity is very important in discussing the definition of LBW. However, I understand the importance of maternal lifestyle and dietary habits in addition to where mothers live, how they spend their time, and what they eat daily. The lifestyle of Caucasian mothers living in Japan may differ from that of those living in the USA. Therefore, the incidence of LBW might be different between Japanese mothers living in Japan and Japanese mothers living in the USA. I believe it is important to mention this fact instead of considering Figure 1 as meaningless. As you know, it might be difficult to discuss and investigate the association between ethnicity and country of residence and perinatal outcomes. As you pointed out, our definition of LBW might be confusing for readers. However, we hope to note the importance of this issue and start the discussion nationwide.

Therefore, I have added the following information to the text and revised the following sentence:

“Ethnicity may be the most important factor for fetal growth. However, although maternal height was correlated with LBW incidence in 19-year-olds categorized by country (Fig. 1) [10], LBW is defined by birth weight and not maternal diversity.” (p2, line 50-52)

“This might mean that Japanese food and lifestyle may control maternal healthcare.” (p2, line 70-71)

In the second section 3, line 117, it is described that “mothers who were born with LBW” should be “mothers who were born as LBW.” 

[Response]

Thank you for your editorial comment. I have revised the sentence.

“Regarding perinatal outcomes, mothers who were born as LBW infants are at a higher risk of delivering LBW or SGA infants [29]. (p4, line 121-122)

Reviewer 2 Report

Comments and Suggestions for Authors

Review LBW and  ethnicity

In this article from Japan the question is rised that a definition for low birth weight of 2500 grams is too non-specific to use it for the DOHaD concept considering separate ethnic groups.

That is a remarkable question from Japan because I found in my literature about HLA-type that in Japan there is no increase in HLA- Dr3 and juvenile diabetes as is in most populations. HLA-Dr4 is uniformly increased  in all population including Japan. For your information we did a study to the relation  of the HLA-type and congenital malformations in the offspring of mothers with juvenile diabetes. Another already known fact is that HLA-Dr2 seems to protect against  juvenile diabetes.But in a group of Hindustani mothers with gestational diabetes ( Asians you call them in your article) there was a high incidence of Dr-2 type. It is known that GDM is higher in Hindustani women.

So there are differences between ethnicity groups in genetic factors for immunity and the aspect of immunity might also be important since this might be related to preeclampsia , a wellknown cause of  LBW.

Maybe in your article you might also involve the HLA-type in relation to LBW.  There is a paper on HLA-G see:

It is found in the article of Journal of Reproductive Immunology

Volume 120, April 2017, Pages 8-14

Journal of Reproductive Immunology

Associations between fetal HLA-G genotype and birth weight and placental weight in a large cohort of pregnant women – Possible implications for HLA diversity

Johanne Emmery aOle B. Christiansen b c, Line Lynge Nilsson a, Mette Dahl a, Peter Skovbo c, Anna Margrethe Møller c, Rudi Steffensen d, Thomas Vauvert F. Hviid a

Otherwise  I like your article . I found to typo’s : line 32 the word were is too much and in line 78 cleated is created??

Author Response

Reviewer 2

In this article from Japan the question is rised that a definition for low birth weight of 2500 grams is too non-specific to use it for the DOHaD concept considering separate ethnic groups.

That is a remarkable question from Japan because I found in my literature about HLA-type that in Japan there is no increase in HLA- Dr3 and juvenile diabetes as is in most populations. HLA-Dr4 is uniformly increased in all population including Japan. For your information we did a study to the relation of the HLA-type and congenital malformations in the offspring of mothers with juvenile diabetes. Another already known fact is that HLA-Dr2 seems to protect against juvenile diabetes. But in a group of Hindustani mothers with gestational diabetes (Asians you call them in your article) there was a high incidence of Dr-2 type. It is known that GDM is higher in Hindustani women. So there are differences between ethnicity groups in genetic factors for immunity and the aspect of immunity might also be important since this might be related to preeclampsia , a well known cause of  LBW.

Maybe in your article you might also involve the HLA-type in relation to LBW.  There is a paper on HLA-G see: Associations between fetal HLA-G genotype and birth weight and placental weight in a large cohort of pregnant women – Possible implications for HLA diversity.

[Response]

Thank you for your editorial comment. I acknowledge the association between HLA type and fetal growth. Therefore, I have cited the report about HLA you recommended and added a discussion about HLA type and fetal growth.

“Moreover, human leukocyte antigen (HLA) type might affect fetal growth. Emmery et al. reported that HLA-G variations were associated with fetal weight and placental weight [18].” (p4, line 71-73)

[18] Emmery, J.; Christiansen, O.B.; Nilsson, L.L.; Dahl, M.; Skovbo, P.; Moller, A.M.; Steffensen, R.; Hviid, T.V.F. Associations between fetal HLA-G genotype and birth weight and placental weight in a large cohort of pregnant women - Possible implications for HLA diversity. J. Reprod. Immunol. 2017, 120, 8-14, doi:10.1016/j.jri.2017.02.002

Round 2

Reviewer 1 Report

Comments and Suggestions for Authors

I appreciate your effort in responding to my concerns. However, I do not think the abstract provides enough information. Can you elaborate a little more, as most readers will only read the abstract? After I pointed out the problem, I was intrigued by why there are two sections, 3 and 4. You brought up the distinction between birth weight < 10th % and weight criteria (2200 vs. 2750 vs. 3000 gm) in describing LBW in section 2. What is your opinion about the best criteria? The content remains highly elusive to me. It is hard to capture the essence of this review article.

-        Line 52: I think diversity should be changed to ethnicity.  

Author Response

Reviewer 1

I appreciate your effort in responding to my concerns. However, I do not think the abstract provides enough information. Can you elaborate a little more, as most readers will only read the abstract?

[Response]

Thank you for your comment. I added the following sentence.

[Abstract]

“We compiled national data from several countries and found that maternal height was negatively correlated with LBW incidence. We discovered the INTERGROWTH-21st chart may not be suitable for the Japanese population as the Japanese birth weight chart differs from the INTERGROWTH-21st chart. Researchers have reported different LBW cutoff values used to assess adverse perinatal outcomes for different countries. However, there is currently no definition of LBW independent of the mother’s country of origin that can be used for predicting the risk of adverse health outcomes. Therefore, the current era of personalized healthcare may be the perfect time to establish a standard definition of LBW which is independent of the mother’s country of origin. Considering the future of healthcare, it seems an apt time to discuss the development of a more meaningful definition of LBW that can be applied across ethnicities. Further research is needed to investigate the cut-off values of LBW in every ethnicity.” (p1, lines 11-22)

After I pointed out the problem, I was intrigued by why there are two sections, 3 and 4. You brought up the distinction between birth weight < 10th % and weight criteria (2200 vs. 2750 vs. 3000 gm) in describing LBW in section 2. What is your opinion about the best criteria? The content remains highly elusive to me. It is hard to capture the essence of this review article.

[Response]

Thank you for your query. I have not understood the suitable cut-off values of LBW in every ethnicity yet. Therefore, I wrote this review and wanted to raise this issue. I added the following sentence.

[Conclusion]

“Therefore, we conclude that further research is needed to investigate the cut-off values of LBW in every ethnicity.” (p4, lines 167-169)

Reviewer 2 Report

Comments and Suggestions for Authors

I have read the new version and I can agree to accept it now. 

Author Response

Reviewer 2

I have read the new version and I can agree to accept it now. 

[Response]

Thank you for your message.